# CONVOLUTIONS ARE COMPETITIVE WITH TRANSFORMERS FOR PROTEIN SEQUENCE PRETRAINING

## ABSTRACT

Pretrained protein sequence language models largely rely on the transformer architecture. However, transformer run-time and memory requirements scale quadratically with sequence length. We investigate the potential of a CNN-based architecture for protein sequence masked language model pretraining and subsequent finetuning. CNNs are competitive on the pretraining task with transformers across several orders of magnitude in parameter size while scaling linearly with sequence length. More importantly, CNNs are competitive with and occasionally superior to transformers across an extensive set of downstream evaluations, including structure prediction, zero-shot mutation effect prediction, and out-of-domain generalization. We also demonstrate strong performance on sequences longer than the positional embeddings allowed in the current state-of-the-art transformer protein masked language models. Finally, we close with a call to disentangle the effects of pretraining task and model architecture when studying pretrained protein sequence models.

## 1 INTRODUCTION

Large pretrained protein language models, largely relying on the attention-based transformer (Vaswani et al., 2017) architecture, have advanced the ability of machine-learning methods to predict protein structure and function from sequence, especially when labeled training data is sparse. Most modern self-supervised protein sequence pretraining combines a transformer model with either an autoregressive likelihood (Madani et al., 2020; 2021; Ferruz et al., 2022; Hesslow et al., 2022) or with the masked language modeling (MLM) task introduced for natural language by BERT (bidirectional encoder representations from transformers) (Devlin et al., 2018). Pretrained transformer protein MLMs contain structural information (Rao et al., 2019; Rives et al., 2021; Chowdhury et al., 2021), encode evolutionary trajectories (Hie et al., 2022a; 2021), are zero-shot predictors of mutation fitness effects (Meier et al., 2021), improve out-of-domain generalization on protein engineering datasets (Dallago et al., 2021), and suggest improved sequences for engineering (Hie et al., 2022b). Protein MLMs are now incorporated into the latest machine-learning methods for detecting signal peptides (Teufel et al., 2021) and predicting intracellular localization(Thumuluri et al., 2022).

One drawback of transformers is that the compute and memory required by the attention layers scale quadratically with input sequence length. In addition, transformer attention is invariant to position, so transformer sequence models include a positional embedding. Depending on the formulation, these embeddings can be difficult to extend past the maximum length seen during training. As a result, some popular pretrained protein transformer models limit the input length during pretraining and inference; for example, ESM has a maximum input length of 1022 residues. Of the 42 million cluster representatives in the March 2020 release of UniRef50 (Suzek et al., 2015), 1.1 million, or 2.6%, are longer than 1022 residues. This includes many proteins of interest, such as the SARS-Cov-2 spike glycoprotein and *Streptococcus pyogenes* CRISPR-associated endonuclease Cas9.

Furthermore, there has been little investigation of how model architecture interacts with pretraining tasks on protein sequences. Transformers can perform the masked language model task on protein sequences, and pretraining improves the performance of transformers on downstream protein structure and property prediction tasks. However, it is important to disentangle pretraining from architectural advances and consider them independently. We seek to do this by investigating the effectiveness of pretrained and naive convolutions for proteins.

We train protein sequence convolutional masked language models on UniRef50, which we refer to as CARP (**C**onvolutional **A**utoencoding **R**epresentations of **P**roteins). Our CARP models are competitive with transformers on the pretraining task, given comparable parameter sizes. The largest CARP, with approximately 640M learnable parameters (CARP-640M) is competitive with the current state-of-the-art transformer protein sequence masked language model, ESM (Rives et al., 2021; Meier et al., 2021) on a variety of downstream prediction tasks, including structure prediction, zero-shot mutation effect prediction, and out-of-domain generalization on biologically-relevant protein engineering datasets. Because CARP scales linearly in computation with the input sequence and does not rely on an input positional embedding, it is straightforward to apply it to sequences longer than the longest sequences in training, which we demonstrate with zero-shot predictions of mutation effects in CRISPR-Cas9. These empirical results demonstrate a need to deepen our understanding of protein sequence pretraining by disentangling the effects of architecture and the pretraining task. Finally, while performance on structure prediction tasks improves as model size and pretraining performance improve, this is not the case for all fitness prediction tasks, demonstrating we also need to deepen our understanding of how pretraining relates to downstream tasks.

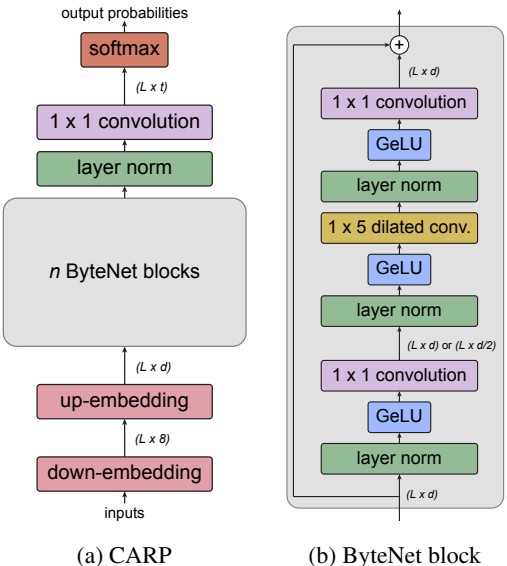

(a) CARP        (b) ByteNet block

Figure 1: The CARP architecture.

## 2 CONVOLUTIONAL PROTEIN SEQUENCE MASK LANGUAGE MODELS

We pretrain CARP using the masked language model (MLM) objective described in Rives et al. (2021). Each sequence is corrupted by changing some tokens to a special mask token or another amino acid token, and the model is tasked with reconstructing the original sequence. Specifically, 15% of tokens from each sequence are randomly selected for supervision. For those 15% of tokens, 80% are replaced by the mask token, 10% are replaced by a randomly-chosen amino acid, and 10% remain unchanged. The model is trained to minimize the cross-entropy loss between its predictions for the selected tokens and the true tokens at those locations. We train on the cluster representatives from the March 2020 release of UniRef50, with approximately 83k sequences held out for validation and another 210k sequences held out for testing, leaving 41.5 million sequences for training.

CARP combines the ByteNet encoder dilated CNN architecture from Kalchbrenner et al. (2016) with simple input embedding and output decoding layers, as shown in Figure 1a. CARP begins with an embedding layer, which maps an input sequence of $L$ tokens $x \in \mathbb{D}^L$ to an 8-dimensional intermediate embedding, followed by a linear mapping into the model dimension $d$: $e_0 \in \mathbb{R}^{L \times d}$. This passes through a stack of $n$ ByteNet dilated CNN blocks Figure 1b with residual connections in between followed by a final layer norm to produce the encoder representation $e_n \in \mathbb{R}^{L \times d}$, and finally a linear decoder maps this to the $L \times t$ logits, where $t$ is the number of possible tokens. The $1 \times 5$ convolution layer in every ByteNet block is dilated and padded to preserve sequence length. Dilation

increases the CNN perceptive field exponentially with the number of layers in order to obtain global context for long input sequences. The CNN dilation rate doubles every layer up to a maximum rate $r$ (for our experiments $r = 128$). The scheme is repeated multiple times in the network, always starting from a dilation rate of 1. While convolution kernels are homogenous across locations, the ByteNet architecture preserves the order of those locations. We found that adding positional embeddings does not improve pretraining performance.

We varied the number of parameters in CARP from approximately 3000 to 640 million by setting the model dimension $d$, setting the encoder hidden dimension $h_e$ to either $d$ or $\frac{d}{2}$, and setting the number of layers. All models are trained with the Adam optimizer, a maximum learning rate of 0.001, a linear warmup for 16,000 steps, and dynamic batching to maximize GPU usage. The largest model, CARP-640M, was trained on 128 32GB Nvidia V100 GPUs for 620,000 updates, or approximately 56 days.

## 3 RELATED WORK

**CNN language models**  CARP's architecture is based on ByteNet (Kalchbrenner et al., 2016), which introduced a dilated convolutional seq2seq framework for neural machine translation. Work in natural language processing (Kalchbrenner et al., 2016; Wu et al., 2019; Tay et al., 2021) hints that pretrained attention-free convolutional neural networks (CNNs) can be competitive with pretrained transformers while scaling linearly with sequence length. CARP directly applies this work to protein sequence pretraining.

**Protein sequence pretraining**  ESM-1b (Rives et al., 2021) is a 650-million-parameter transformer protein masked language model trained on the March 2018 release of UniRef50 (Suzek et al., 2007). ESM-1v (Meier et al., 2021) uses the same transformer architecture, but is optimized for mutation-effect prediction by training on UniRef90 instead of UniRef50. TAPE (Rao et al., 2019) is a smaller transformer protein masked language model trained on protein domains from Pfam (Mistry et al., 2021) instead of the full protein sequences found in UniRef. ProtTrans (Elnaggar et al., 2021) explores the use of different transformer architectures language modeling tasks and larger datasets. The most comparable ProtTrans models are ProtBERT-UniRef100 and ProtBERT-BFD, which are 420-million-parameter transformers protein masked language models trained on UniRef100 and BFD (Steinegger and Söding, 2018; Steinegger et al., 2019), respectively. ProteinBERT (Brandes et al., 2021) introduces a global attention mechanism and an additional functional annotation prediction task during pretraining. Rao et al. (2021) extends the transformer masked language model scheme to multiple sequence alignments.

**Convolutional models of protein sequences**  Shin et al. (2021) train autoregressive convolutional models on protein families, but do not attempt to train a single model over the breadth of known protein sequence diversity. Lu et al. (2020) use a small convolutional encoder for a noise-contrastive pretraining task on proteins, but do not give it global context or make the model autoencoding. Bileschi et al. (2022) use a similar convolutional architecture to our model to learn functional annotations for unaligned protein sequences. However, their task is not autoencoding, and they do not consider performance on downstream tasks.

By combining a denoising autoencoding task with a dilated CNN architecture, we begin to disentangle the effect of pretraining task from the effect of model architecture.

## 4 PRETRAINING PERFORMANCE

Our largest model, CARP-640M, has a test loss of 2.02, comparable to ESM-1b, which has 650 million parameters and a loss of 1.96 on its test set. Note that ESM-1b was trained and tested on an earlier version of UniRef50 with different train/test splits than CARP or our ESM models. (Throughout, ESM-1b refers specifically to the 650-million parameter transformer trained on the March 2018 UniRef50 release and described in Rives et al. (2021), while ESM refers to our small transformer masked language models based off of the ESM-1b architecture. Likewise, CARP refers to any ByteNet masked language model, while CARP-X refers to the model with approximately X

parameters.) Hyperparameters for different-sized versions of CARP and ESM are found in Tables A1 and A2, respectively.

For comparison, we also trained transformer models with comparable numbers of parameters using the ESM-1b architecture described in Rives et al. (2021) on our UniRef50 dataset. As shown in Figure 2a, CARP's performance on the pretraining task is comparable to ESM's across several orders of magnitude of variation in the number of parameters when using the same pretraining dataset. Figure 2b shows MLM loss by length for CARP-640M and ESM-1b on their respective test sets, smoothed with a window of 30 in the length dimension. For both models, the pretraining loss improves quickly until the sequence length reaches about 500, and then slowly thereafter. The maximum input length for ESM-1b is 1022, but we calculate losses for CARP-640M for sequences with up to 4096 residues. These results show that convolutions can perform protein sequence masked language modeling comparably to tranformers without suffering from a quadratic dependence between runtime and sequence length.

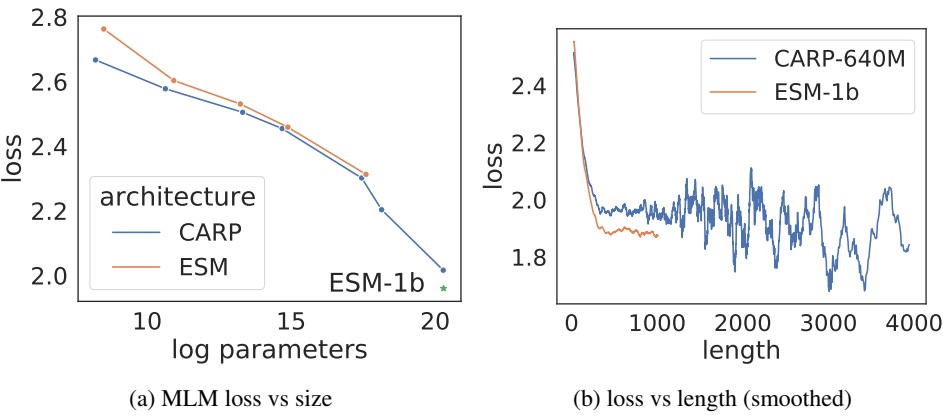

(a) MLM loss vs size          (b) loss vs length (smoothed)

Figure 2: Comparisons between CARP and the ESM-1b transformer.

## 5   DOWNSTREAM TASKS

One goal of protein MLMs is to encode information in their output representation or model weights that improves performance on downstream prediction tasks. Downstream evaluation can be zero-shot (without access to labels for further training), the pretrained model can be frozen and a small neural network decoder can be trained to predict labels from the pretrained model's output representations (pt-fr), or the new decoder and pretrained model can be finetuned together (pt-ft). We use the output from the final layer norm in Figure 1a as the output representation. Unless otherwise noted, the new decoder consists of a learned attention that converts the output from $L \times d$ to $d$ followed by a 2-layer neural network with hidden size $d$. For tasks with labels, we evaluate both pt-fr and pt-ft and compare to ESM-1b or ESM-1v. We finetune models with a maximum learning rate of 0.0001, a linear warmup over 1000 steps, and early stopping based on the validation set. Finetuning was performed on one 32 GB V100; depending on the task, finetuning took between several minutes to 48 hours. Where relevant, we also compare the CARP architecture with randomly-initialized weights (na-fr and na-ft), linear ridge regression, and the small CNN described in Dallago et al. (2021) and Shanehsazzadeh et al. (2020).

### 5.1   PROTEIN STRUCTURE

One of the most striking successes of protein MLMs is their ability to encode structural information without access to structural labels during pretraining. We evaluate CARP-640M's ability to encode structural information through 3 tasks:

1. **Remote contact prediction** asks a model to predict whether the $C_\beta$ atoms of two residues separated by at least 24 residues in the primary structure are within 8 Angstroms of other in the three-dimensional structure. We train on the trRosetta (Yang et al., 2020) training set

and evaluate the precision of the top $L$ predictions on the CAMEO hard (Haas et al., 2018) and CASP13-FM (Shrestha et al., 2019) test sets. For contact prediction, we downsample CARP embeddings to 128 dimensions, perform an outer product to produce 2-dimensional embeddings, and then pass that to a 24-layer dilated residual CNN based on the trRosetta architecture. This is the same as the procedure used by ESM-1b.

2. **Remote homology detection** asks a model to detect structural similarity across distantly-related sequences. We evaluate accuracy on the fold-level holdout set from TAPE.

3. **3-class secondary structure prediction** asks a model to predict whether each residue in a protein is part of a helix, strand, or other. We use the training and validation sets from TAPE and evaluate accuracy on the CB513 test set. For this task, we train a neural network consisting of two CNN layers, an LSTM, and a linear head on top of the pretrained model, as described in Rives et al. (2021).

As shown in Table 1, pretraining improves performance for structure prediction tasks, and CARP-640M is competitive with ESM-1b. These results show that pretrained convolutions learn structural information from single sequences, just as pretrained transformers do.

Table 1: Structure prediction tasks. Values for ESM-1b are taken from Rives et al. (2021). Uncertainties are standard deviations on 3 replicates with different weight initializations.

| Method | Model | Task | | | |
|--------|-------|-----------|--------|-----------------|---------------------|
| | | CASP-13 FM | CAMEO | remote homology | secondary structure |
| pt-fr | CARP-640M | 23.7 | 42.0 | 0.24±0.008 | **0.83**±0.001 |
| | ESM-1b | **28.2** | **44.4** | - | **0.82** |
| pt-ft | CARP-640M | - | - | 0.28±0.008 | **0.83**±0.001 |
| | ESM-1b | - | - | **0.33** | - |
| na-fr | CARP-640M | 9.7 | 12.6 | 0.09±0.02 | 0.65±0.02 |
| na-ft | CARP-640M | - | - | 0.09 ± 0.02 | 0.71±0.0005 |

## 5.2 ZERO-SHOT MUTATION EFFECT PREDICTION

Large language models can predict experimental measurements of protein function without further training on sequence-fitness measurements or sets of evolutionarily-related sequences Hie et al. (2022a); Meier et al. (2021). Following Meier et al. (2021), we score CARP-640M on 41 deep mutational scanning datasets originally compiled by Riesselman et al. (2018). These datasets measure the effects of thousands of mutations or combinations of mutations to a parent sequence. Details are described in Section A.2.

Figure A1 compares zero-shot performance for CARP-640M, ESM-1b, ESM-1v, position-specific scoring matrices (PSSM), and ProtBert-BFD. ESM-1v results are for an ensemble of five transformers. Averaged across the 41 datasets, CARP-640M has a Spearman correlation of 0.49, compared to 0.46 for ESM-1b, 0.51 for ESM-1v, 0.46 for PSSM, and 0.43 for ProtBERT-BFD. CARP-640M outperforms ESM-1b on 22 out of 41 datasets, ESM-1v on 18 out of 41 datasets, PSSM on 26 out of 41 datasets, and ProtBERT-BFD on 25 out of 41 datasets.

Meier et al. (2021) found that using the full UniProt sequences instead of only the sequence of the mutated domain results in better zero-shot predictions. However, this is not always possible with ESM-1x, as some UniProt sequences for these proteins are longer than 1022 residues. As a further proof of concept, we made zero-shot predictions for the effects of mutations in Cas9 from *Streptococcus pyogenes* (Spencer and Zhang, 2017), which is 1368 residues long, and obtain a Spearman correlation of 0.26. These results show that pretrained convolutions can make zero-shot predictions of protein mutation effects on fitness, including on sequences longer than allowed by ESM-1x.

## 5.3 OUT-OF-DOMAIN FITNESS PREDICTION

Another motivation for pretrained protein sequence models is that they may be able to generalize after fine-tuning in ways that are helpful for protein engineering. For example, a protein engineer may want to train a model on single mutants and make predictions for sequences with multiple mutations, or train a model that is accurate for sequences with fitness greater than what is seen in the training set. Here, we evaluate CARP-640M on tasks from the following landscapes from FLIP (Dallago et al., 2021).

1. AAV (Table 2): Adeno-associated virus (AAV) capsid proteins are responsible for helping the virus integrate a DNA payload into a target cell (Vandenberghe et al., 2009), and there is great interest in engineering versions of these proteins for gene therapy (Büning et al., 2015; Barnes et al., 2019). Bryant et al. (2021) measure a rich mutational screening landscape of different VP-1 AAV proteins.

2. GB1 (Table 3): GB1 is the binding domain of protein G, an immunoglobulin binding protein found in Streptococcal bacteria. In their original study, Wu et al. (2016) measured the fitness of 149,361 out of 160, 000 possible combinations of mutations at 4 positions.

For each landscape, we evaluate several tasks, including:

- x-vs-many: Train on sequences with up to x mutations and test on the remainder of the landscape.
- mut-des: Train on sequences sampled from mutagenesis libraries and test on sequences designed by machine-learning models (AAV only).
- low-vs-high: Train on sequences with fitnesses below the wild-type and test on sequences with fitnesses above the wild-type.

We compare results to ESM-1b and the same decoder neural network, linear ridge regression, and a small CNN.

Table 2: Performance on the FLIP AAV tasks. Values for models other than CARP-640M are taken from Dallago et al. (2021). Uncertainties for ESM-1b and CNN are standard deviations over 10 random seeds. Uncertainties for CARP-640M are standard deviations over 3 random seeds. Dallago et al. (2021) do not provide uncertainties for the mut-des task because of the computational cost.

| Method | Model | Task | | | | |
|---|---|---|---|---|---|---|
| | | 1-vs-many | 2-vs-many | 7-vs-many | mut-des | low-vs-high |
| pt-fr | CARP-640M | 0.31±0.18 | 0.51±0.18 | 0.58±0.14 | 0.75±0.08 | 0.25±0.09 |
| | ESM-1b | 0.03±0.11 | 0.61±0.04 | 0.65±0.01 | 0.76 | **0.38**±0.01 |
| pt-ft | CARP-640M | **0.73**±0.05 | **0.81**±0.03 | **0.77**±0.03 | **0.85**±0.003 | 0.19±0.08 |
| na-fr | CARP-640M | 0.48±0.07 | 0.50±0.05 | 0.60±0.05. | 0.76±0.02 | 0.21±0.02 |
| | ESM-1b | 0.18±0.01 | 0.20±0.03 | 0.38±0.04 | 0.56 | 0.06±0.01 |
| na-fr | CARP-640M | 0.04±0.12. | 0.50±0.43 | 0.38±0.37 | 0.84±0.01 | 0.24±0.21 |
| baseline | ridge | 0.22 | 0.03 | 0.65 | 0.68 | 0.12 |
| | CNN | 0.35±0.11 | 0.58±0.09 | 0.73±0.004 | 0.71 | 0.28±0.02 |

In general, pretraining improves CARP-640M's performance on these tasks, and fine-tuning the entire model outperforms freezing the pretrained weights. Comparisons to the baselines show that pretraining is most helpful when generalizing from single mutants to multiple. When not fine-tuning all the way through, there is little benefit from pretraining, and on some tasks pretraining hurts performance. CARP-640M outperforms ESM-1b on generalizing from few mutations to more, but ESM-1b is better at generalizing from a low-fitness training to higher-fitness sequences. These results show that pretrained convolutions help generalization to types of sequence variation not seen during training. On GB1, finetuning ESM-1b end-to-end instead of freezing the pretrained weights hurts its performance, while CARP-640M benefits from full finetuning. We do not finetune ESM-1b end-to-end on other tasks because of the computational cost. In addition, CARP-640M provides

better representations without pretraining than ESM-1b on all the AAV tasks and 2 of the 4 GB1 tasks, showing that the architecture alone also influences generalization.

Table 3: Performance (Spearman correlation) on the FLIP GB1 tasks. Values for models other than CARP-640M and ESM-1b with full finetuning are taken from FLIP. Uncertainties for ESM-1b frozen and CNN are standard deviations over 10 random seeds. Uncertainties for CARP-640M and ESM-1b with full finetuning are standard deviations over 3 random seeds.

| Method | Model | Task | | | |
|---|---|---|---|---|---|
| | | 1-vs-many | 2-vs-many | 3-vs-many | low-vs-high |
| pt-fr | CARP-640M | 0.15±0.18 | 0.18±0.23 | 0.62±0.06 | 0.12±0.03 |
| | ESM-1b | **0.29**±0.02 | 0.47±0.05 | 0.79±0.01 | **0.53**±0.03 |
| pt-ft | CARP-640M | 0.19±0.26 | **0.73**±0.03 | **0.87**±0.004 | 0.43±0.04 |
| | ESM-1b | 0.11±0.11 | 0.67±0.07 | 0.66±0.18 | 0.42±0.09 |
| na-fr | CARP-640M | 0.03±0.03 | 0.07±0.17 | 0.71±0.03 | 0.35±0.03 |
| | ESM-1b | 0.12±0.01 | 0.21±0.01 | 0.52±0.01 | 0.32±0.03 |
| na-ft | CARP-640M | 0.11±0.07 | 0.38±0.26 | 0.68±0.33 | 0.23±0.26 |
| | ESM-1b | 0.05±0.28 | 0.14±0.13 | 0.10±0.13 | -0.04±0.09 |
| baseline | ridge | 0.28 | 0.59 | 0.76 | 0.34 |
| | CNN | 0.15±0.09 | 0.39±0.04 | 0.81±0.004 | 0.47±0.01 |

## 5.4 IN-DOMAIN PROPERTY PREDICTION

Finally, we consider fitness-prediction tasks that do not require difficult biological generalization (Table 4). We evaluate on three sequence-fitness regression tasks:

1. **Fluorescence** requires the model to predict the effect of one or more mutations on the brightness of green fluorescent protein. The data was originally collected by Sarkisyan et al. (2016). We use the data splits provided in TAPE.

2. **Stability** requires the model to predict a small protein's resistance to protease degradation. The data was originally collected by Rocklin et al. (2017). We use the data splits provided in TAPE.

3. **Meltome-mixed** requires the model to predict the melting temperature of a variety of proteins from across the domains of life. The data was originally collected by Jarzab et al. (2020). We use the cluster representatives and data splits provided in FLIP.

in addition to two intrinsically-disordered region (IDR) function classification tasks taken from Zarin et al. (2021). For the IDR datasets, we use MMseqs2 (Steinegger and Söding, 2017) to cluster sequences to 50% identity and then randomly assign clusters to training, validation, or testing.

1. **Cdc28 binding** requires the model to predict whether an IDR is a target of Cdc28.
2. **Mitochondria targeting** requires the model to predict whether an IDR targets its protein for transport into the mitochondria.

In general, while pretraining improves CARP-640M's performance on these tasks, neither of the large pretrained models consistently out-perform the baseline models on these tasks. Almost all the models perform very well on the IDR tasks, indicating that performance is saturating on these tasks. Nevertheless, CARP-640M is generally comparable to ESM-1b, showing that once again pretrained convolutions are comparable to pretrained attention.

## 5.5 EFFECT OF MODEL SIZE AND PRETRAINING PERFORMANCE ON DOWNSTREAM PERFORMANCE

To investigate the effects of model size and pretraining performance on downstream performance, we finetune pretraining checkpoints for CARP-600k, CARP-76M, and CARP-640M on secondary structure, remote homology, and the FLIP GB1, AAV, and meltome tasks. In each of these experiments,

Table 4: Performance on in-domain tasks. For fluorescence, stability, and meltome, values reported are Spearman correlation. For the IDR tasks, values reported are area under the reciever operating curve. Values for ESM-1b on fluorescence and stability are taken from Rives et al. (2021). Values for baselines on fluorescence and stability are taken from FLIP. Uncertainties for ESM-1b and CNN are standard deviations over 10 random seeds. Uncertainties for CARP-640M are standard deviations over 3 random seeds. We do not calculate uncertainties on meltome due to the computational cost.

| Method | Model | Task | | | | |
| --- | --- | --- | --- | --- | --- | --- |
| | | fluorescence | stability | meltome | Cdc28 | mito. |
| pt-fr | CARP-640M | 0.58±0.02 | 0.62±0.03 | 0.54 | 0.84±0.01 | 0.86±0.02 |
| | ESM-1b | - | - | **0.67**±0.01 | **0.91**±0.004 | **0.90**±0.01 |
| pt-ft | CARP-640M | **0.68**±0.002 | **0.72**±0.01 | 0.53 | 0.88±0.02 | 0.89±0.004 |
| | ESM-1b | **0.68** | 0.71 | - | 0.89±0.01 | 0.88±0.01 |
| na-fr | CARP-640M | 0.62±0.01 | 0.52±0.17 | 0.29 | 0.84±0.01 | 0.86±0.01 |
| | ESM-1b | - | - | 0.45±0.03 | 0.88±0.01 | 0.84±0.03 |
| na-ft | CARP-640M | 0.58±0.07 | 0.65±0.05 | 0.30 | 0.79±0.03 | 0.87±0.01 |
| | ESM-1b | - | - | - | 0.83±0.02 | 0.85±0.02 |
| | ridge | **0.68** | 0.48 | 0.17 | 0.52 | 0.53 |
| | CNN | 0.67 | 0.51 | 0.34±0.01 | 0.84±0.02 | 0.87±0.02 |

we initialize the prediction head with the same weights across all model checkpoints of the same size. Figures 3a and A3a show that structure prediction improves smoothly as the model size increases and the model is pretrained longer. This confirms that, as for transformers, pretraining imparts structural information to CNNs. However, Figures 3b, A3b, and A3c shows that this relationship does not exist for the out-of-domain FLIP tasks. In many cases, a small amount of pretraining is sufficient to outperform the naive baseline, and further pretraining has an unpredictable and often negative effect on performance. Using the FLIP meltome task as an example of an in-domain class, Figure A3d shows that performance generally improves as CARP is pretrained, but the pretraining effect saturates, and CARP-76M outperforms CARP-640M.

Figure 4a shows that the average zero-shot performance improves with both model size and pretraining performance. However, this is not the case for every individual dataset within DeepSequence. Figure 4b shows a case where zero-shot performance peaks and then declines as CARP is pretrained. The Spearman correlation between the pretrain loss and zero-shot Spearman correlation range from 1 (monotonic increase in zero-shot performance with more pretraining) and -0.9, as shown in Figure A2. The average over the DeepSequence datasets is 0.40 for CARP-640M, 0.48 for CARP-76M, and 0.23 for CARP-600k. Although CARP-640M has better overall zero-shot performance than CARP-76M, CARP-76M more consistently improves with more pretraining than CARP-640M. The heterogeneity in the relationship between pretraining performance and zero-shot performance suggests that many but not all zero-shot tasks in DeepSequence are strongly determined by structural stability.

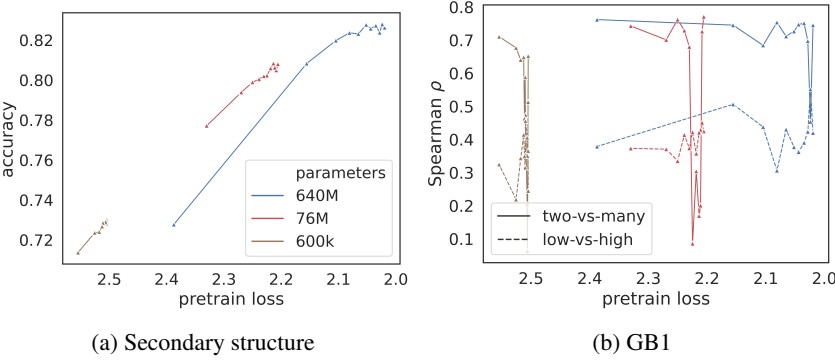

(a) Secondary structure      (b) GB1

Figure 3: Effect of model size and checkpoint pretrain loss on downstream performance.

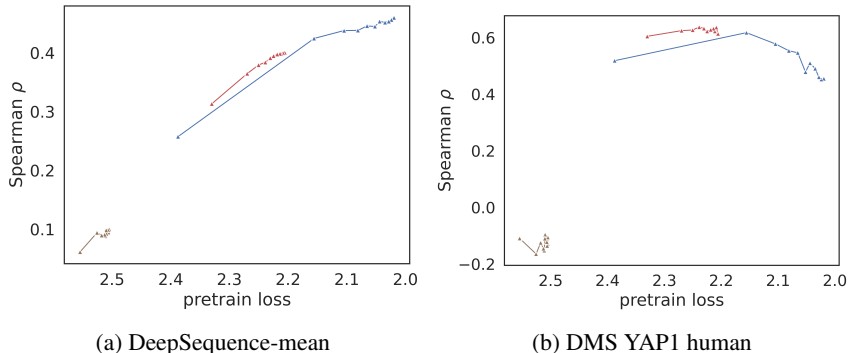

(a) DeepSequence-mean            (b) DMS YAP1 human

Figure 4: Effect of model size and checkpoint pretrain loss on zero-shot performance.

# 6 CONCLUSIONS

We have shown that convolutions can be comparable to or superior to transformers on both the MLM pretraining task and a variety of downstream protein sequence modeling tasks, and that convolutions, like transformers, benefit from pretraining. Furthermore, without pretraining, convolutions and transformers perform differently on downstream tasks, showing the important of disentangling pretraining and architecture. Unlike transformers, convolutions scale linearly with input sequence length, which becomes important when modeling long protein sequences. Work in natural language processing has also shown that convolutions can require fewer FLOPs of compute than transformers, even for short sequences (Tay et al., 2021). In addition, while we use standard dilated convolutions, there are more efficient convolution variants designed for sequence modeling (Wu et al., 2019) that may further improve model speed.

**Limitations** However, convolutions may not be competitive with transformers on tasks where a cross- or self-attention inductive bias is explicitly needed or desired for interpretability. For example, it is possible to extract structural contact maps from pretrained transformer self-attention matrices (Rao et al., 2020), and self-attention matrices contain information about binding sites (Vig et al., 2020) – convolutions lack an obvious equivalent. In addition, it is more natural to extend attention-based models to predict protein-protein interaction sites. The transformer's quadratic dependence on sequence length can also be ameliorated with approximate attention methods (Child et al., 2019; Beltagy et al., 2020; Kitaev et al., 2020; Tay et al., 2020a; Wang et al., 2020; Zaheer et al., 2020; Katharopoulos et al., 2020; Choromanski et al., 2020a), but the choice of approximation matters for performance and the best method is not always clear *a priori* (Tay et al., 2020b). On proteins, Choromanski et al. (2020a) and Choromanski et al. (2020b) show that Performer approximate attention can perform well for autoregressive and masked protein language models, respectively, while ProteinBERT combines a fast global attention mechanism with masked language and functional annotation prediction pretraining (Brandes et al., 2021).

**Outlook** Currently, pretrained protein language models are tightly-coupled to the transformer architecture, and the effects of the pretraining task can be conflated with the effects of the pretrained architecture. Our pretrained convolutional models may provide complementary inductive biases those found in pretrained transformer models, making them useful alternatives for practitioners. Unfortunately, we also find that, while masked language model pretraining is very effective for imparting models with structural knowledge, the relationship between model size, pretrain loss, and downstream performance is more fraught for out-of-domain protein engineering tasks, indicating the need for more effective pretraining tasks. We hope that this work is the first step in investigating the independent and interaction effects of pretraining and architecture for protein sequence modeling. While we evaluate the effects of masked language model pretraining, transformers have also been used for autoregressive language model pretraining (Madani et al., 2020) and pairwise masked language modeling (He et al., 2021), and combining structural information (Mansoor et al., 2021; Zhang et al., 2022; McPartlon et al., 2022; Hsu et al., 2022; Chen et al., 2022; Wang et al., 2022) or functional annotations (Brandes et al., 2021) offers further directions for protein pretraining tasks.

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

# A   APPENDIX

## A.1   HYPERPARAMETERS FOR PRETRAINED MODELS OF DIFFERENT SIZES

All models are trained for 2 weeks on 1-8 32GB V100 GPUs with dynamic batching. Table A1 summarizes the hyperparameters for CARP. Table A2 summarizes the hyperparameters for ESM.

Table A1: CARP model hyperparameters. Max tokens is the maximum number of tokens per GPU per batch during training.

| Model | Parameters | Layers | $d$ | $d_{\mathrm{MLP}}$ | Max tokens | GPUs |
|---|---|---|---|---|---|---|
| CARP-640M | 643M | 56 | 1280 | 1280 | 11000 | $128 \times 32\mathrm{GB}$ V100 |
| CARP-76M | 75.7M | 32 | 1024 | 512 | 60000 | $16 \times 32$ GB V100 |
| CARP-38M | 37.9M | 16 | 1024 | 512 | 40000 | $8 \times 32$ GB V100 |
| CARP-24M | 23.9M | 16 | 256 | 128 | 400000 | $2 \times 32$ GB V100 |
| CARP-600k | 608k | 16 | 128 | 64 | 600000 | $1 \times 32$ GB V100 |
| CARP-40k | 415k | 16 | 32 | 16 | 600000 | $1 \times 32$ GB V100 |
| CARP-4k | 3670 | 16 | 8 | 4 | 600000 | $1 \times 16$ GB V100 |

Table A2: ESM model hyperparameters.

| Parameters | Layers | Heads | $d$ | $d_{\mathrm{MLP}}$ | GPUs |
|---|---|---|---|---|---|
| 86.5M | 12 | 12 | 768 | 3972 | $8 \times 32$ GB V100 |
| 44.0M | 6 | 12 | 768 | 3072 | $8 \times 32$ GB V100 |
| 2.92M | 6 | 8 | 192 | 768 | $2 \times 32$ GB V100 |
| 561k | 4 | 6 | 96 | 384 | $1 \times 32$ GB V100 |
| 41.5k | 4 | 4 | 24 | 96 | $1 \times 32$ GB V100 |
| 4890 | 2 | 4 | 4 | 16 | $1 \times 32$ GB V100 |

## A.2   ZERO-SHOT FITNESS PREDICTION

We score sequences by masking every mutated position and computing the log odds ratio between the mutated and wild-type residues at each mutated position, assuming an additive model when a sequence contains multiple mutations:

$$\sum_{p \in P} \log p(x_P^{\mathrm{mt}} | x_{\backslash P}^{\mathrm{wt}}) - \log p(x_P^{\mathrm{wt}} | x_{\backslash P}^{\mathrm{wt}}) \tag{1}$$

where $P$ indicates the mutated positions.

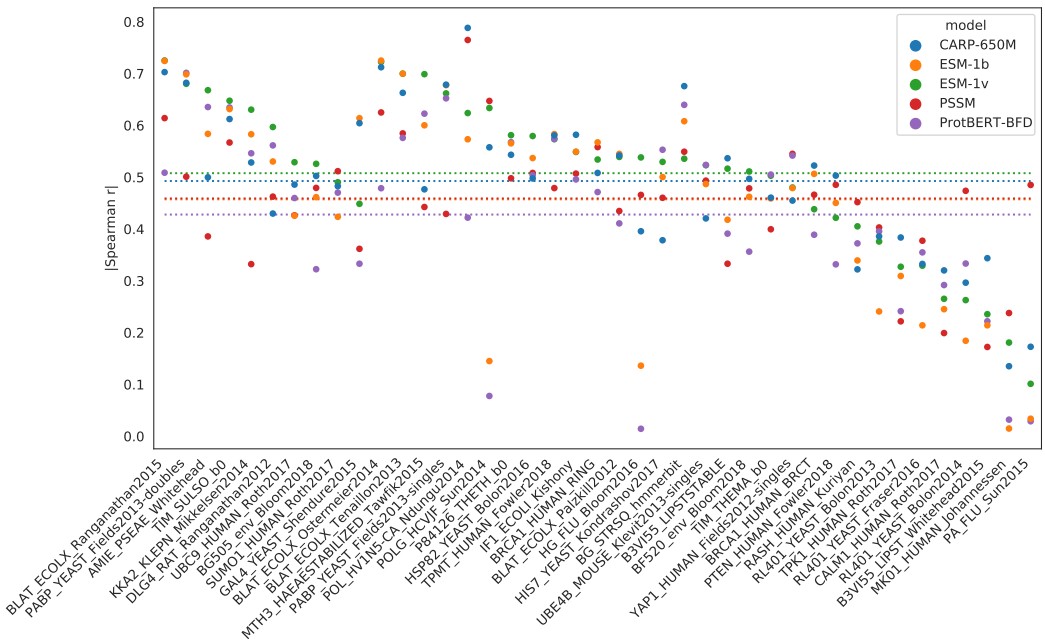

Figure A1: Zero-shot protein fitness prediction. Comparison across 41 deep mutational scanning datasets from DeepSequence. Points are Spearman correlation on each dataset. Horizontal lines show the average Spearman correlation across the datasets. Values for ESM-1b, ESM-1v, PSSM, and ProtBERT-BFD are taken from Meier et al. (2021).

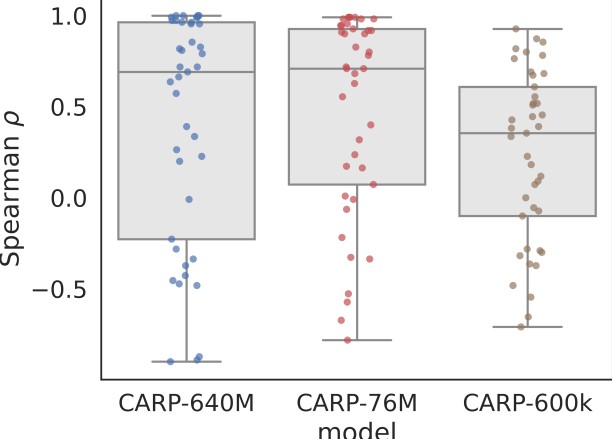

Figure A2: Spearman correlation between pretrained model checkpoint loss and zero-shot Spearman correlation across 41 deep mutational scanning datasets from DeepSequence.

## A.3 Pretrain performance vs downstream performance

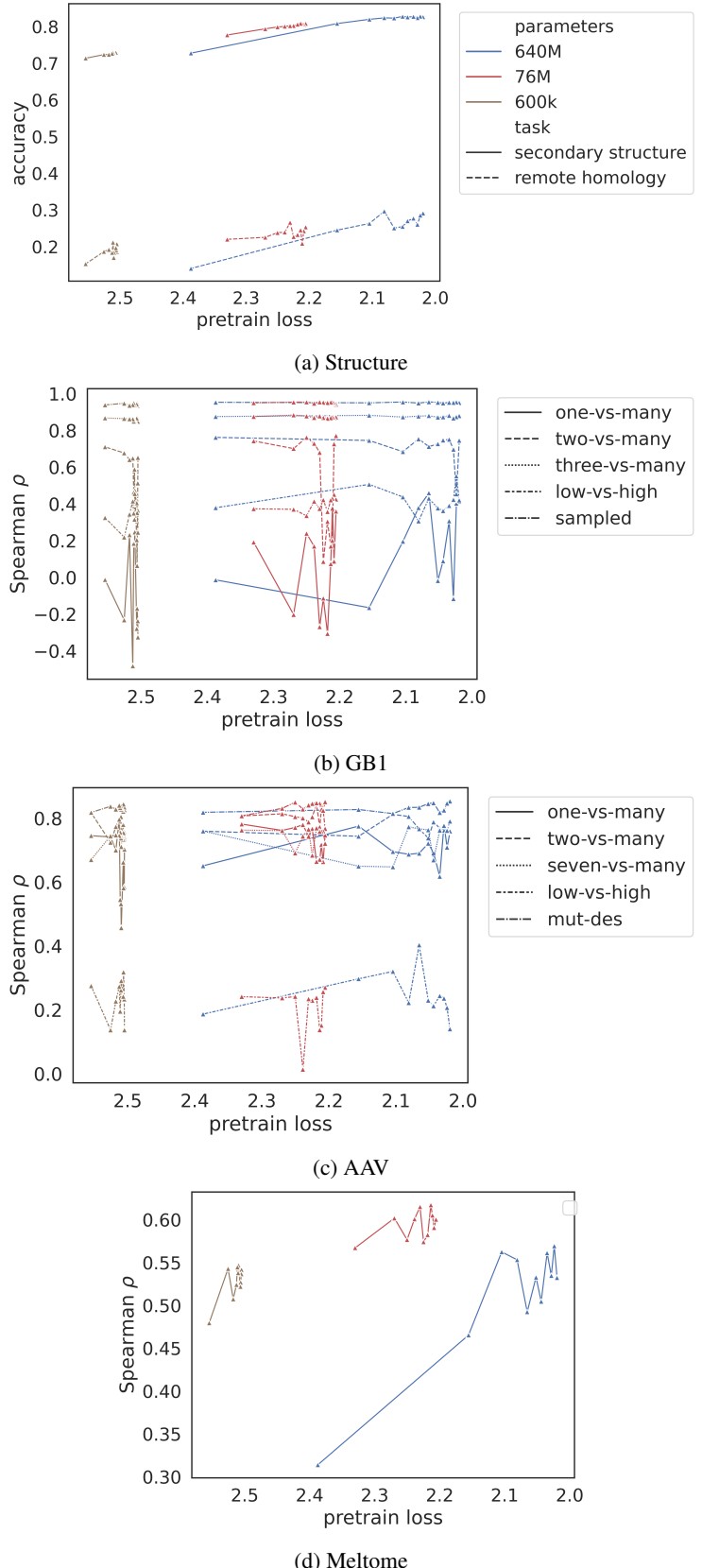

(a) Structure

(b) GB1

(c) AAV

(d) Meltome

Figure A3: Downstream performance vs pretrain loss for secondary structure, remote homology, and FLIP tasks.

