# OpenReview forum: "Convolutions are competitive with transformers for protein sequence pretraining"
_ICLR.cc/2023/Conference — Submitted to ICLR 2023_

### Official Review · Reviewer_79qs · 2022-10-23

**Confidence:** 4
**Correctness:** 3
**Technical Novelty And Significance:** 3
**Empirical Novelty And Significance:** 3
**Recommendation:** 6

**Clarity, Quality, Novelty And Reproducibility:**

the presentation is clear.

although the model and the experiments are not very original, the argument that the effect of pre-training should be decoupled from the choice of model architecture is reasonable.

**Strength And Weaknesses:**

*Strength*

1. The presentation is clear, and the decoupling claim is reasonable.

2. Empirical results are comprehensive, and downstream tasks contain testing various aspects of a pre-trained model.


*Weaknesses*

1. there are many ways to handle longer sequences during inference for transformers, for example, from the perspective of positional embeddings, one can use a set of fixed embeddings or recently proposed rotary embeddings.

2. it wasn't immediately clear to me whether positional embeddings were used for convolutional neural networks. Since convolution kernels are homogenous across locations, I imagine that positional embeddings would be helpful.

3. it always puzzles me whether long-distance attention is learnt in transformers, and I understand that people have designed downstream tasks that require long-distance attention to perform well, but what if the realised tasks themselves could be solved with a decent accuracy using only short-distance attention? I was wondering if there exists work on checking the biases in the downstream tasks?

**Summary Of The Paper:**

the submission compared convolution neural networks of the same building block and the transformer models on the impact of masked pre-training on downstream tasks including secondary structure predictions and protein design tasks. Results show that there isn't a significant difference between convolutional neural networks and transformer models on downstream tasks. Based on the empirical evidence, the submission argues that the effect of pre-training should be decoupled from the architercture.

**Summary Of The Review:**

I would vote for weak acceptance of this paper so that researchers in the field are aware of the decent performance of convolutional neural networks and their argument on decoupling.

---

> ### Author Response · Authors · 2022-11-08
> **response to reviewer 79qs**
>
> Thank you for the kind and thorough review. Here are replies to specific concerns.
>
> >there are many ways to handle longer sequences during inference for transformers, for example, from the perspective of positional embeddings, one can use a set of fixed embeddings or recently proposed rotary embeddings.
>
> This is true, and we will clarify that this is a weakness of the embeddings used in ESM, not positional embeddings in general.
>
> >it wasn't immediately clear to me whether positional embeddings were used for convolutional neural networks. Since convolution kernels are homogenous across locations, I imagine that positional embeddings would be helpful.
>
> We do not use positional embeddings. While convolution kernels are homogenous across locations, the ByteNet architecture preserves the order of those locations. Early on in this project, we did test using positional embeddings and found that they did not help on the pretrain task. We will clarify this in the paper.
>
> >it always puzzles me whether long-distance attention is learnt in transformers, and I understand that people have designed downstream tasks that require long-distance attention to perform well, but what if the realised tasks themselves could be solved with a decent accuracy using only short-distance attention? I was wondering if there exists work on checking the biases in the downstream tasks?
>
> The structure prediction tasks probably do explicitly require long-distance attention. There’s some unpublished work showing that if you perturb inputs to ESM, the resulting changes in predictions recapitulate structural contacts. However, we are not aware of any work on the engineering-related downstream tasks.

---

### Official Review · Reviewer_JHic · 2022-10-24

**Confidence:** 4
**Clarity, Quality, Novelty And Reproducibility:** The explanations are good enough for …
**Correctness:** 2
**Technical Novelty And Significance:** 2
**Empirical Novelty And Significance:** 2
**Recommendation:** 3

**Strength And Weaknesses:**

Authors proposed the alternative approaches to the Transformer using the CNN architecture. Standard dilated convolutions are involved to further boost the speed.

Compared to the ESM-1b paper, the results are not good;sometimes showing inferior results. In some fields, the proposed CNN-based model is showing better accuracy in some cases; at the same time it is showing inferior results for other cases. I am not clearly convinced with the authors’ arguments saying that CNNs are alternative to the Transformers.
Also, there is no result for longer sequences; while this was the initial question for this draft.


**Summary Of The Paper:**

Protein sequence language models are mostly based on the Transformer architecture; while the complexity of it increases exponentially. Thus, this paper studies the use of CNNs instead of Transformer for masked language model pre-training and finetuning. Authors proposed the CARP model for protein sequence pre-training. CNN exhibits the linear complexity according to the length of sequences. More importantly, in some downstream evaluations for estimating the structure prediction and zero-shot mutation effect prediction and for OOD generalization, CNN performs better than the Transformer.


**Summary Of The Review:**

The motivation for this paper is good; while I do not think the current draft is effectively offering the evidence for better speed and accuracy. Thus, I recommend reject for this paper.

---

> ### Author Response · Authors · 2022-11-08
> **Response to reviewer JHic**
>
> Thank you for the review. Here are responses to your critiques.
>
> >Compared to the ESM-1b paper, the results are not good;sometimes showing inferior results. In some fields, the proposed CNN-based model is showing better accuracy in some cases; at the same time it is showing inferior results for other cases. I am not clearly convinced with the authors’ arguments saying that CNNs are alternative to the Transformers.
>
> >I do not think the current draft is effectively offering the evidence for better speed and accuracy
>
> We do not claim that CARP-640M is better than ESM-1b – simply that it is comparable for most downstream tasks: better in some, worse in others, as also stated in your review. Our primary conclusions are that researchers should consider CNNs alongside transformers for these tasks, and that neither architecture combined with masked language model pretraining performs well for out-of-domain fitness prediction. We believe (and the other reviewers concur) that our results support these conclusions. If you disagree that the paper presents sufficient evidence for these, could you explain more or provide an example of evidence that you would find more convincing?
>
>
> >Also, there is no result for longer sequences; while this was the initial question for this draft.
>
> In section 5.2, we provide a proof of concept by doing zero-shot learning on a cas9 deep mutational scan. As stated in the introduction and discussion, longer sequences are one motivation for this work. Others include showing that researchers should not over-index on the transformer architecture, and that for OOD tasks, it is likely that we need both new architectures and new pretraining tasks. We will revise the paper to highlight these motivations.

---

> > ### Comment · Reviewer_JHic · 2022-11-21
> > **Respond to authors**
> >
> > I still cannot understand the argument of the paper. If some algorithm works better in some cases; while works worse in some other cases, why is it deserved to be published in TOP CONFERENCE? For example, classical machine learning algorithms such as random forest sometimes beats CNNs in some data or in some applications; however researchers do not publish papers focusing on that specific phenomenon. Why the fact that CNNs "sometimes" work better than the Transformer is worth publishing? Please guide me if I am missing something.
> >
> > Also, regarding the longer sequences, I request authors more concrete plans for the revision.

---

> > > ### Author Response · Authors · 2022-11-22
> > > **Response to JHic**
> > >
> > > We respectfully disagree with the reviewer's view that a paper only deserves publication if it presents an algorithmic advance that achieves top performance across the board. We believe the point of publications at conferences is to advance discussion and standards in the research community: while we agree that better performing methods are one way to do this, pointing out limitations with and broadening current research directions can also contribute. As it stands, most current research in protein language models assumes that Transformers are superior, even though there has been no rigorous comparative benchmarking, simply because these are the standard models used in NLP. Our work provides the first systematic comparison, and challenges the notion that we can directly transfer pre-built models in language to proteins, a totally different domain. Because of these long-standing conceptions, the majority of work for proteins has focused on Transformers, with comparatively little research done on CNNs - our work confirms that CNNs is a viable strategy, and we view our paper as enabling a new direction of research with CNN-based architectures for future researchers, as opposed to leaving this direction unexplored because Transformers are assumed to be superior.

---

> ### Author Response · Authors · 2022-11-14
> **Please respond to help us improve our paper :-)**
>
> Thank you again for the time you spent reviewing our paper. Would it be possible to respond to our comment early enough that we could try to address any points that you think would help us improve the paper or your rating?

---

### Official Review · Reviewer_i3wu · 2022-10-25

**Confidence:** 4
**Correctness:** 3
**Technical Novelty And Significance:** 3
**Empirical Novelty And Significance:** 3
**Recommendation:** 6

**Clarity, Quality, Novelty And Reproducibility:**

The paper is clear and well written. The finding that convolutional nets can outperform transformers on out-of-domain evaluations seems surprising and perhaps novel.

**Strength And Weaknesses:**

Strengths:
* The paper gives an interesting account of current work on protein sequence language modeling. This might be useful for researchers unfamiliar with these applications.
* The observation that the authors’ CNN (CARP) obtains nearly the same loss during pretraining as a transformer (ESM) is quite interesting and surprising. (Figure 2)
* The paper makes the interesting observation that model size and pre-training don’t always improve downstream tasks.
* The out-of-domain results seem to strongly favor CARP. This could be interesting, since it shows that for some tasks CNNs might be generalizing better in this application space.

Weaknesses:
* The paper says “The **primary** drawback of transformers is that the compute and memory required by the attention layers scale quadratically with input sequence length“. This is too strong a statement. Quadratic dependency on the length is, at best, **one** of the drawbacks, but probably not the primary one. There are many versions of transformers that don’t have quadratic dependence on the length of the sequence and can outperform convolution approaches, e.g. Big Bird (Zaheer et al, 2020, cited in this paper), Performer (Choromanski et al, 2020, also cited in this paper).
* The paper claims that “[positional] encodings can be difficult to extend past the maximum length seen during training”. Where are the limits coming however on the input length during training? If the length during training is capped artificially because of memory requirements, then this limitation can be removed again by using an architecture like Big Bird, and by simply further training Big Bird on longer input sequences after the initial pre-training.
* Figure 2b is potentially misleading, since it’s comparing MLM loss on different test sets (March 2018 UniRef50 for ESM-1b vs March 2020 UniRef50 for CARP). Why didn’t the authors evaluate ESM-1b on the March 2020 UniRef50 test set? The MLM loss for higher sequence lengths is also very noisy, indicating that there are perhaps very few sequences of high lengths?
For example the CNN results in Table 1 are markedly lower than the transformer based ESM-1b baseline. The authors show this table to show that CNNs are not at too large a disadvantage compared to transformers, but the gap still seems large. ESM-1b also wins on zero-shot evaluations, although CNNs seem very close in this case.
* The paper draws a distinction between convolutional nets and transformers, but it should be conceptually simple to create a hybrid architecture, by adding an attention mechanism and positional embeddings in the convolution. It could be interesting to examine this in future work.


**Summary Of The Paper:**

The paper reports a set of experimental results based on protein sequence masked language modeling using a convolutional architecture, which they name "CARP", comparing to ESM-1b, a previously published transformer based model. The authors show that
  * CARP performs similarly to ESM-1b in terms of MLM loss, for different orders of magnitudes of parameter counts,
  * CARP is sometimes better than ESM-1b on some downstream tasks, specifically when the evaluation set is out of domain, but it's worse on tasks that involve structure prediction of proteins,
  * CARP might generalize better to sequence lengths longer than the ones seen in training.

The authors make some progress towards disentangling the effect of pretraining data and transformer architecture on protein sequence pretraining applications. They show that in some cases pretraining more actually hurts performance on downstream tasks.

**Summary Of The Review:**

The paper experiments with a convolutional architecture for protein sequence pretraining and shows that in some downstream tasks it can outpeform a recently published transformer model. The improvements seem to mostly be on out-of-domain tasks (Table 2).

It's a bit unclear how significant these out-of-domain improvements are, since they are only compared to the ESM-1b baseline, which seems to do terribly on some of these tasks.

Still the discussion is quite interesting to read and it could conceivably lead to a better architecture in future work, by combining the ESM and CARP architectures perhaps, or by finding a way to improve the transformer architecture to do better in out-of-domain evaluations.

---

> ### Author Response · Authors · 2022-11-08
> **response to i3wu**
>
> Thank you for the kind and thorough review. We will clarify some claims and writing based on your suggestions. Specific responses are below.
>
> >The paper says “The primary drawback of transformers is that the compute and memory required by the attention layers scale quadratically with input sequence length“. This is too strong a statement. Quadratic dependency on the length is, at best, one of the drawbacks, but probably not the primary one. There are many versions of transformers that don’t have quadratic dependence on the length of the sequence and can outperform convolution approaches, e.g. Big Bird (Zaheer et al, 2020, cited in this paper), Performer (Choromanski et al, 2020, also cited in this paper).
>
> Thank you for pointing this out. We will clarify in the introduction that this is one drawback of the original transformer attention formulation.
>
> >The paper claims that “[positional] encodings can be difficult to extend past the maximum length seen during training”. Where are the limits coming however on the input length during training? If the length during training is capped artificially because of memory requirements, then this limitation can be removed again by using an architecture like Big Bird, and by simply further training Big Bird on longer input sequences after the initial pre-training.
>
> This is a weakness of the positional embeddings used in ESM. There do exist positional embeddings that do not have this property. We agree that a two-step procedure can be used to get around memory limitations: one benefit of our model is that this is not required.
>
> >Figure 2b is potentially misleading, since it’s comparing MLM loss on different test sets (March 2018 UniRef50 for ESM-1b vs March 2020 UniRef50 for CARP). Why didn’t the authors evaluate ESM-1b on the March 2020 UniRef50 test set? The MLM loss for higher sequence lengths is also very noisy, indicating that there are perhaps very few sequences of high lengths? For example the CNN results in Table 1 are markedly lower than the transformer based ESM-1b baseline. The authors show this table to show that CNNs are not at too large a disadvantage compared to transformers, but the gap still seems large.
>
> We do not evaluate ESM-1b on the March 2020 UniRef50 test set because this test set includes sequences that are in ESM-1b’s training set. There are indeed not very many sequences in UniRef at each length greater than about 1000 residues, leading to the noisy results.
> CARP-640M does indeed underperform ESM-1b on structure prediction. However, the hyperparameters for ESM-1b were chosen by optimizing performance for structure prediction, whereas we do not tune the hyperparameters for CARP. We suspect that tuning hyperparameters in the same way to produce a CARP-1b-640M could close the gap here.
>
> >ESM-1b also wins on zero-shot evaluations, although CNNs seem very close in this case.
>
> ESM-1b does not win on zero-shot evaluations. CARP-640M has an average Spearman of 0.49 on DeepSequence, compared to 0.46 for ESM-1b. The ESM-1v ensemble does outperform CARP-640M with a Spearman of 0.51, but that’s trained on UniRef90, which they show is better for variant prediction, and furthermore is an ensemble of predictions from 5 pretrained models, each with 650M parameters.
>
> >The paper draws a distinction between convolutional nets and transformers, but it should be conceptually simple to create a hybrid architecture, by adding an attention mechanism and positional embeddings in the convolution. It could be interesting to examine this in future work.
>
> We agree! In general, an autoML approach that learns to combine attention and convolutional blocks should do very well here.

---

### Author Response · Authors · 2022-11-08
**General response to reviewers**

We thank all reviewers for taking the time to review our paper. In general, we appreciate the reviewers' positive reception of the paper. The reviewers generally expressed that our finding that CNNs are effective as pretrained language models is of broad interest to the community. As some reviewers point out, this is a surprising finding given that transformers are dominant in this space. Based on this, we are excited about the impact and discussion that our findings will generate in the protein machine learning community: as the reviewers suggest, our results point to many future experiments, such as combining aspects of transformers and CNNs.

We plan to clarify our claims and writing based upon the feedback from the reviewers. Several of the reviewers pointed out that our claims about the weaknesses of transformers are overstated, and there are now methods to overcome memory limitations in these architectures. We agree with these critiques, and will clarify our claims.

We do believe that reviewer JHic may have some misconceptions about the claims we are making in this paper when they say that our results are "not good" because we show "inferior results" on some tasks. To be clear, we have never claimed that CNNs should generally be used instead of transformers, only that they should be studied alongside transformers for protein modeling tasks. Given the diversity of tasks evaluated, it is unlikely that any single model will be the best at all of them. Supporting this claim, our paper shows that CNNs are better at some tasks, and worse than others. We will clarify the language in the manuscript to make this claim more explicit.

We will reply to individual reviews to address specific concerns.

---

### Author Response · Authors · 2022-11-18
**Revision summary**

We have uploaded a review with a few additional experiments and clarifications requested by the reviewers. We hope that the reviewers will carefully consider the new manuscript and our replies when deciding their final decisions.

The additional experiments consist of end-to-end finetuning for ESM-1b on the GB1 tasks (Table 3). We find that finetuning end-to-end hurts ESM-1b performance compared to freezing the pretrained weights, while the opposite is true for CARP-640M.

---

### Decision · Program_Chairs · 2023-01-20

**Decision:**

Reject

**Justification For Why Not Higher Score:**

N/A

**Justification For Why Not Lower Score:**

N/A

**Metareview: Summary, Strengths And Weaknesses:**

The main claim of the paper is that despite the Transformer is used as a dominate solution for sequence modeling in protein settings, CNNs are also a viable solution as well and CNNs enjoy a linear complexity in computation which is very suitable for long sequence modeling tasks in bioinformatics. The reviewers and the authors had a good discussion about the paper and the authors are very responsive to the questions that the reviewers posted. After summarizing the reviews and the revisions, I believe the paper does not meet the acceptance standard now. The main reason I think are two folded. First, I believe mentioning that using CNNs is a viable solution is a good contribution, however, it’s a somewhat obvious thing to the research community. In NLP, people using CNNs as the replacement for LSTMs in machine translation and many other applications as well. It isn’t a particular surprising conclusion or a very novel usage. Second, perhaps more importantly, the comparison in experiments may point out CNNs’ comparable results, but it does not offer quite a lot of insights of why it’s doing better when comparing with transforms, especially in the protein modeling setting. As a paper which holds such claims, giving more insights I believe is a very important thing. I therefore recommend not to accept the paper.